

# Genetic dissection of grain iron concentration in hexaploid wheat (*Triticum aestivum* L.) using a genome-wide association analysis method

Jiansheng Wang[1,2,3,*], Xia Shi[2,*], Zhengfu Zhou[2], Maomao Qin[2], Yahuan Wang[2], Wenxu Li[2], Pan Yang[2], Zhengqing Wu[2] and Zhensheng Lei[1,2]

[1] College of Chemistry and Environment Engineering, Pingdingshan University, Pingdingshan, Henan Province, CHINA
[2] Wheat Research Institute, Henan Academy of Agricultural Sciences, Zhengzhou, Henan province, CHINA
[3] Henan Key Laboratory of Germplasm Innovation and Utilization of Eco-economic Woody Plant, Pingdingshan, Henan province, CHINA
* These authors contributed equally to this work.

## ABSTRACT

Iron (Fe) is an essential micronutrient of the body. Low concentrations of bioavailable Fe in staple food result in micronutrient malnutrition. Wheat (*Triticum aestivum* L.) is the most important global food crop and thus has become an important source of iron for people. Breeding nutritious wheat with high grain-Fe content has become an effective means of alleviating malnutrition. Understanding the genetic basis of micronutrient concentration in wheat grains may provide useful information for breeding for high Fe varieties through marker-assisted selection (MAS). Hence, in the present study, genome-wide association studies (GWAS) were conducted for grain Fe. An association panel of 207 accessions was genotyped using a 660K SNP array and phenotyped for grain Fe content at three locations. The genotypic and phenotypic data obtained thus were used for GWAS. A total of 911 SNPs were significantly associated with grain Fe concentrations. These SNPs were distributed on all 21 wheat chromosomes, and each SNP explained 5.79–25.31% of the phenotypic variations. Notably, the two significant SNPs (AX-108912427 and AX-94729264) not only have a more significant effect on grain Fe concentration but also have the reliability under the different environments. Furthermore, candidate genes potentially associated with grain Fe concentration were predicted, and 10 candidate genes were identified. These candidate genes were related to transport, translocation, remobilization, and accumulationof ironin wheat plants. These findings will not only help in better understanding the molecular basis of Fe accumulation in grains, but also provide elite wheat germplasms to develop Fe-rich wheat varieties through breeding.

Corresponding authors
Zhengfu Zhou, zhouzf215@163.com
Zhensheng Lei, zhenshenglei@126.com

## INTRODUCTION

Micronutrient malnutrition is caused by a lack of important micronutrients such as iron (Fe). Fe is a critical micronutrient and has several important functions in the body; for example, it plays a central role in the transportation of blood oxygen, and reduced Fe intake can lead to impaired growth and behavioral problems (*Welch & Graham, 1999*). It is also called 'hidden hunger' due to the appearance of undesirable symptoms with few visible warning signs that could impair the mental and physical development of humans and generate long-term effects on human health (*Alderman, Hoddinott & Kinsey, 2006*). Microelement deficiencies are common in developed and developing countries, and have become a major global health concern. Experts have estimated that one-third of the world population is at risk of Fe deficiency (*Alloway, 2009*). Women of child-bearing age and children are more prone to microelement deficiencies because they have greater micronutrient needs (*Grzeszczak, Kwiatkowski & Kosik-Bogacka, 2020*). Therefore, adequate intake of essential minerals is important to eliminating 'hidden hunger'. Because the majority of the world's population depends on a few staple crops, such as wheat, rice, and maize, biofortification of these food crops seems to be a promising approach to address dietary mineral deficiencies (*Khush et al., 2012*). Several attempts have been made towards mineral improvement in plants, of which traditional breeding and genetic engineering techniques have been considered to be the most feasible and cost-effective approaches (*Bouis, 2003*).

Wheat (*Triticum aestivum* L.), one of the most important staple food crops in the world, contributes more than 50% of the diet and up to 60% of daily intakes of Fe and Zn in several developing countries (*Cakmak, Pfeiffer & McClafferty, 2010*). Because wheat has many advantages such as wide agronomic flexibility and ease of storage, billions of people depend on wheat to fulfill their nutritional prerequisites. Hence, a sustainable way to protect the population from mineral deficiencies is to improve the nutritional quality of wheat through breeding and selection of varieties with naturally high mineral content. This requires a better understanding of the genetic basis of mineral element accumulation in wheat grains.

Inheritance of micronutrients is quantitative in nature. Linkage mapping and association analysis are useful methods in identifying QTLs for mineral elements. Linkage mapping generally involves specific populations such as recombinant inbred lines (RILs), doubled haploid (DH) lines, $F_{2:3}$ families, and backcross populations ($BC_xF_y$) to identify QTLs of target traits (*Groos et al., 2007*; *Wang et al., 2011*). However, these populations can only be used to identify QTLs for a limited number of traits. Establishing these population is also time-consuming and labor-intensive. Compared with linkage mapping, genome-wide association studies (GWAS) based on linkage disequilibrium (LD) capitalizes on historical recombination, and thus could identify more QTLs related to complex traits at a higher mapping resolution (*Falconer & MacKay, 1996*). Therefore, GWAS has become an important approach in QTL mapping for important and complex agronomic traits (*Gao et al., 2016*). Various studies have been carried out to identify QTLs associated with wheat grain Fe (*Arora et al., 2019*; *Alomari et al., 2018*; *Crespo-Herrera*

*et al., 2017*; *Cu et al., 2020*; *Gorafi et al., 2016*; *Kumar et al., 2018*; *Peleg et al., 2009*; *Rathan et al., 2021*; *Roshanzamir, Kordenaeej & Bostani, 2013*; *Shi et al., 2013*; *Srinivasa et al., 2014*; *Tiwari et al., 2016*; *Velu et al., 2017*; *Wang et al., 2021a*; *Wang et al., 2021b*). Furthermore, *Uauy et al. (2006)* have reported a NAC gene (*NAM-B1*) that associated with increased grain Fe content.

In GWAS analysis, the low density of molecular markers could cause loss of linkages between markers and loci of target traits. High-density linkage maps are important to high-resolution QTL mapping and identification of candidate genes. Hence, plentiful molecular markers are imperative to construct wheat saturate genetic maps, and to significantly improve the efficiency of QTL mapping in GWAS. Compared with other types of molecular markers, single-nucleotide polymorphisms (SNPs) have many advantages such as the most abundant DNA sequence variation present in plant genomes, are virtually unlimited, evenly distributed along the genome, bi-allelic, and co-dominant (*Akpinar, Lucas & Budak, 2017*), making SNPs ideal molecular markers in GWAS analysis. With the development of new sequencing technologies, methods of increasing the number of SNPs have been developed in wheat. Particularly, recently developed SNP gene chips have provided larger numbers of SNP markers. To date, SNP chips have been widely used in QTL analysis for important agronomic traits (*Gao et al., 2016*; *Cui et al., 2017*).

Elevation of essential mineral concentrations in grains is an effective strategy for improving the nutritional value of wheat to prevent micronutrient malnutrition. In the present study, we performed associative analysis with 207 wheat accessions from eight countries using 660K SNP chips to identify QTLs for the concentration of grain Fe. This study was conducted to identify QTLs and candidate genes related to grain Fe content that may be useful for wheat biofortification.

## MATERIALS AND METHODS

### Plant materials and field trials

The association panel used in the present study contained 207 wheat diverse accessions, comprising 194 accessions from the different wheat planting regions of China including Henan Province, Shannxi Province, Jiangsu Province, Shandong Province, Sichuan Province, Hebei Province, Shanxi Province, Beijing City, Anhui Province, Hubei Province, Guizhou Province, Yunnan Province, Ningxia Province, and Heilongjiang Province and 13 accessions from seven other countries, including Russia, France, Mexico, Japan, Australia, Bulgaria, and Romania. All of the accessions were grown at Yuanyang (YY, E 113°37′, N 35°12′), Kaifeng (KF, E 114°30′, N 34°80′) and Shangqiu (SQ, E 115°65′, N 34°45′) in Henan Province during the 2016 cropping seasons. Soil Fe content of the three experiment locations were measured, and the mean Fe content of soil for Shangqiu, Yuanyang, and Kaifeng was 26.7, 26.4, and 26.1 mg/kg, respectively. Field trials were conducted in randomized complete blocks with three replicates at all locations. Each plot contained three 2-m rows spaced 20 cm apart. Agronomic management followed local practices. At maturity, wheat seeds were harvested separately for each accession under the different planted locations. In every location, one sample of seeds was collected for each replicated field plot, and a total of three samples were obtained for each accession.

## Determination of grain Fe concentration

Grains were harvested from each accession of the association population in the Kaifeng, Shangqiu and Yuanyang environments when they were mature. The following method was used for the analysis of grain Fe concentration (*Zarcinas, Cartwright & Spencer, 1987*). First, grains were washed thoroughly with purified water three times to remove soil and dirt. Grains were dried in an oven at 70 °C for 72 h and ground into a fine powder that could pass through a 1-mm screen. Then, 50 micrograms of powdered samples from each sample were microwave digested with 5 mL nitric acid ($HNO_3$) and 2 mL hydrogen peroxide ($H_2O_2$) in polypropylene tubes using a microwave accelerate reaction system (CEM, Charlotte, NC, USA). Subsequently, Fe concentrations in the solutions were measured by a flame Atomic Absorption Spectrometer (AAS) (model 1100; Perkin-Elmer, Seer Green, UK). Meanwhile, blank samples and standard samples were added each time for reference. All of the results represent the average of three replications.

## Genotyping and quality control

The samples consisting of 207 wheat accessions were genotyped using the Affymetrix 660K SNP array comprising 630,517 SNPs and performed by Capital Bio-Corporation, Beijing, China (http://www.capitalbiotech.com/), following the manufacturer's protocol as described by *Akhunov, Nicolet & Dvorak (2009)*. To ensure the quality of genotyping data, sample call rate, SNP call rate, minor allele frequency (MAF), and Hardy-Weinberg equilibrium (HWE) were analyzed. In addition, accuracy was checked for SNP clustering, and manual adjustments were made for incorrectly clustered SNPs. The SNPs with a minor allele frequency (MAF) <0.05 and missing data over 20% were excluded from further data analysis. The physical positions of SNP markers from 660K SNP arrays were obtained from the International Wheat Genome Sequencing Consortium website (IWGSC, http://www.wheatgenome.org/). Finally, a total of 224,706 high-quality SNP markers were used for GWAS analysis.

## Genome-wide association analysis and haplotype analysis

In the present study, a total of 224,706 SNPs with a minor allele frequency ≥5% were used for the GWAS. Associations between genotypic and phenotypic data were performed with MLM using the kinship matrix by GAPIT package in R software (*Lipka et al., 2012*). The P value determining whether a SNP marker was associated with Fe concentration and $R^2$ were used to evaluate the magnitude of the MTA (significant marker-trait association) effects. GWAS was conducted for wheat grain Fe concentration at YY, KF, and SQ environments. Common haplotype patterns were assessed in Haploview version4.2 and haplotype blocks were defined with the confidence interval method.

## Candidate gene identification

In order to identify the candidate genes for SNP flanking regions, the flanking DNA sequences corresponding to the SNP markers significantly associated with Fe concentration were used in BLAST searches against the reported common wheat reference genome sequence in NCBI databases (https://blast.ncbi.nlm.nih.gov/Blast.cgi). The high

**Table 1 Descriptive statistics for grain Fe concentrations in GWAS population and soil Fe content in three environments.**

| Environment | Max[a] (mg/kg) | Min[b] (mg/kg) | Mean[c] (mg/kg) | SD[d] | Skew[e] | Kurtosis[f] | Mean content for soil (mg/kg)[g] |
|---|---|---|---|---|---|---|---|
| Kaifeng (KF) | 250.62 | 1.33 | 99.57[h] | 64.50 | 0.5945 | −0.6106 | 26.1 |
| Shangqiu(SQ) | 158.27 | 20.23 | 74.00[i] | 21.69 | 0.3568 | 0.8754 | 26.7 |
| Yuanyang (YY) | 143.78 | 28.15 | 68.12[j] | 22.38 | 0.9795 | 0.8321 | 26.4 |
| Mean | 184.22 | 16.57 | 80.56 | 36.19 | 0.6436 | 0.3656 | 26.4 |

Note:
[a]The maximum value of grain Fe concentration. [b]The minimum value of grain Fe concentration. [c]The mean value of grain Fe concentration. [d]Standard deviation. [e]Kurtosis refer to a measure of the 'tailedness' of the probability distribution of a real-valued random variable. [f]Skewness refer to a measure of the asymmetry of the probability distribution of a real-valued random variable about its mean. [g]The mean value of soil Fe content for differentenvironments. [h, i, j]refer to the significant differences in the means of wheat grain-Fe content for different environments ($P < 0.05$) obtained by ANOVA analysis.

confidence gene list of wheat was also obtained from the International Wheat Genome Sequence Consortium (IWGSC) website (https://wheat-urgi.versailles.inra.fr/) and used to identify possible candidate genes for each identified loci. The annotation of the candidate genes was accomplished with InterProScan (http://www.ebi.ac.uk/interpro/scan.html). The transcript and the corresponding annotation of candidate genes were obtained from the website of IWGSC. For the loci that no candidates found in its mapping interval, the gene close to the peak SNP of the loci was assigned as the candidate.

# RESULTS

## Phenotypic variation for grain Fe concentration in wheat populations

Grain Fe concentration was tested for 207 genotypes in three environments (Table S1). The range and mean of grain Fe concentration in the accessions are presented in Table 1 and Fig. 1. Grain Fe concentration of accessions for the different environments are also depicted in Fig. 1. Grain Fe concentration varied from 1.33–250.62 mg/kg at Kaifeng, 20.23–158.27 mg/kg at Shangqiu, and 28.15–143.78 mg/kg at Yuanyang. Mean Fe concentration was 99.57 mg/kg at Kaifeng, 74.00 mg/kg at Shangqiu, and 68.12 mg/kg at Yuanyang. Grain Fe concentration varied from 16.57 to 184.22 mg/kg (mean: 80.56 mg/kg) among the three locations. In Kaifeng, the highest Fe concentration was recorded in genotype KH438 (250.62 mg/kg), followed by KH445 (245.85 mg/kg), and KH242 (237.74 mg/kg), whereas the lowest Fe concentration was recorded in genotype KH378 (1.33 mg/kg) followed by KH324 (4.67 mg/kg). In Shangqiu, the highest Fe concentration was recorded in genotype SH257 (158.27 mg/kg), followed by SH373 (134.24 mg/kg), and SH272 (133.85 mg/kg), but minimum Fe content was recorded in genotype SH418 (20.23 mg/kg), followed by SH349 (20.24 mg/kg). However, in Yuanyang, the highest Fe concentration was recorded in genotype AH418 (143.78 mg/kg), followed by AH429 (133.76 mg/kg), and AH413 (131.38 mg/kg), while the lowest Fe content was recorded in genotype AH249 (28.15 mg/kg), followed by AH240 (29.59 mg/kg). The mean Fe concentrations of all of the accessions from three locations could be described in decreasing order as follows: Kaifeng (99.57 mg/kg) > Shangqiu (74.00 mg/kg) > Yuanyang (68.12 mg/kg). On the basis of the above statistical analysis, wheat accessions with high Fe concentrations were recommended for breeding cultivars and selected as donors for Fe mineral biofortification in the future. Population distributions of GWAS accessions for
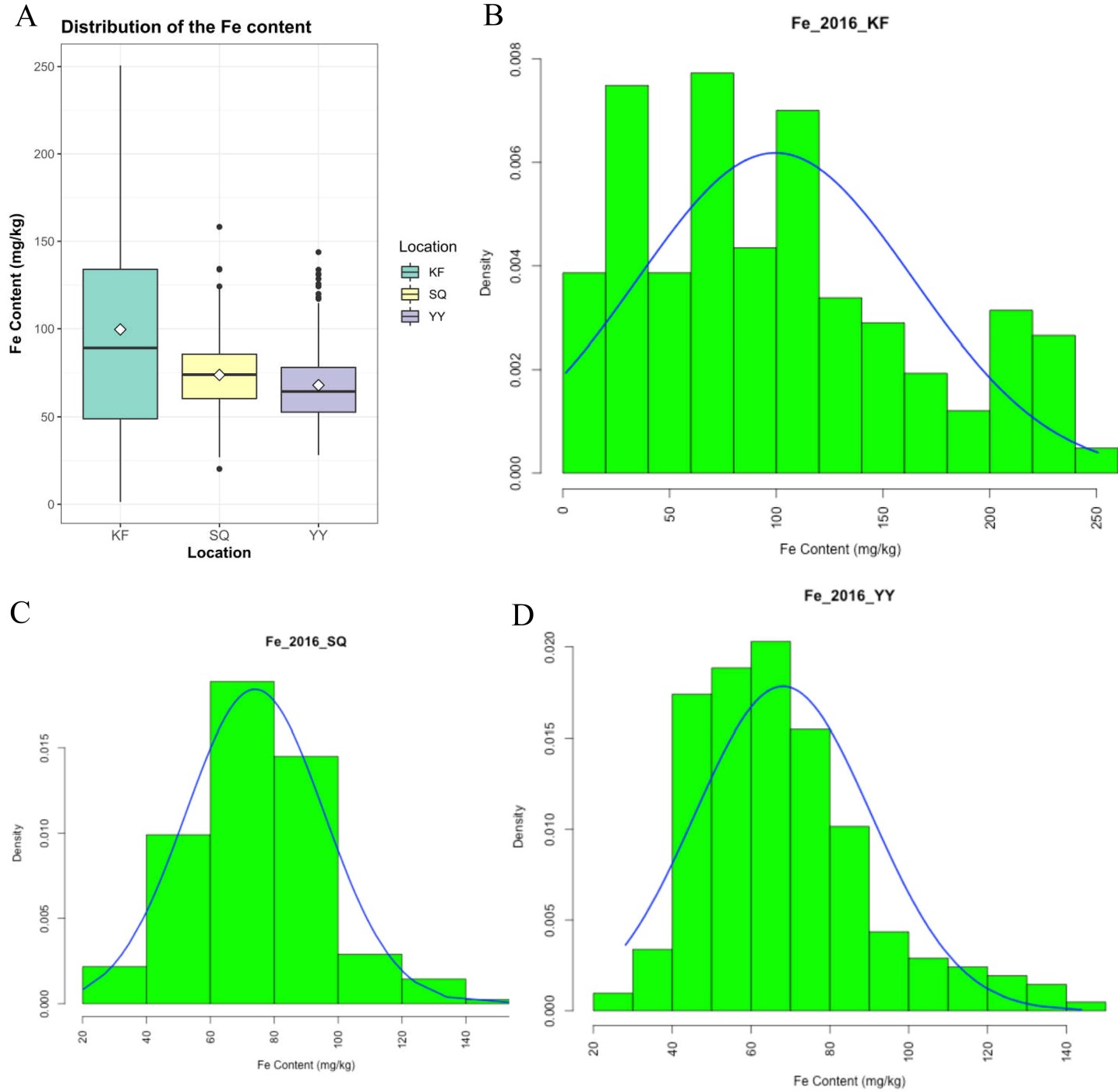

**Figure 1 Distribution of grain Fe concentrations in the wheat association analysis population.** (A) Box plot for grain Fe concentration in three environments (Shangqiu, Yuanyang and Kaifeng). (B–D) Distribution of grain Fe concentration for the wheat association analysis population in Kaifeng (KF), Shangqiu (SQ), Yuanyang (YY) environment, respectively. 

grain Fe concentrations were continuous and exhibited a wide range of values for each location (Fig. 1), which showed that the inheritance of grain Fe was consistent with the quantitative trait.

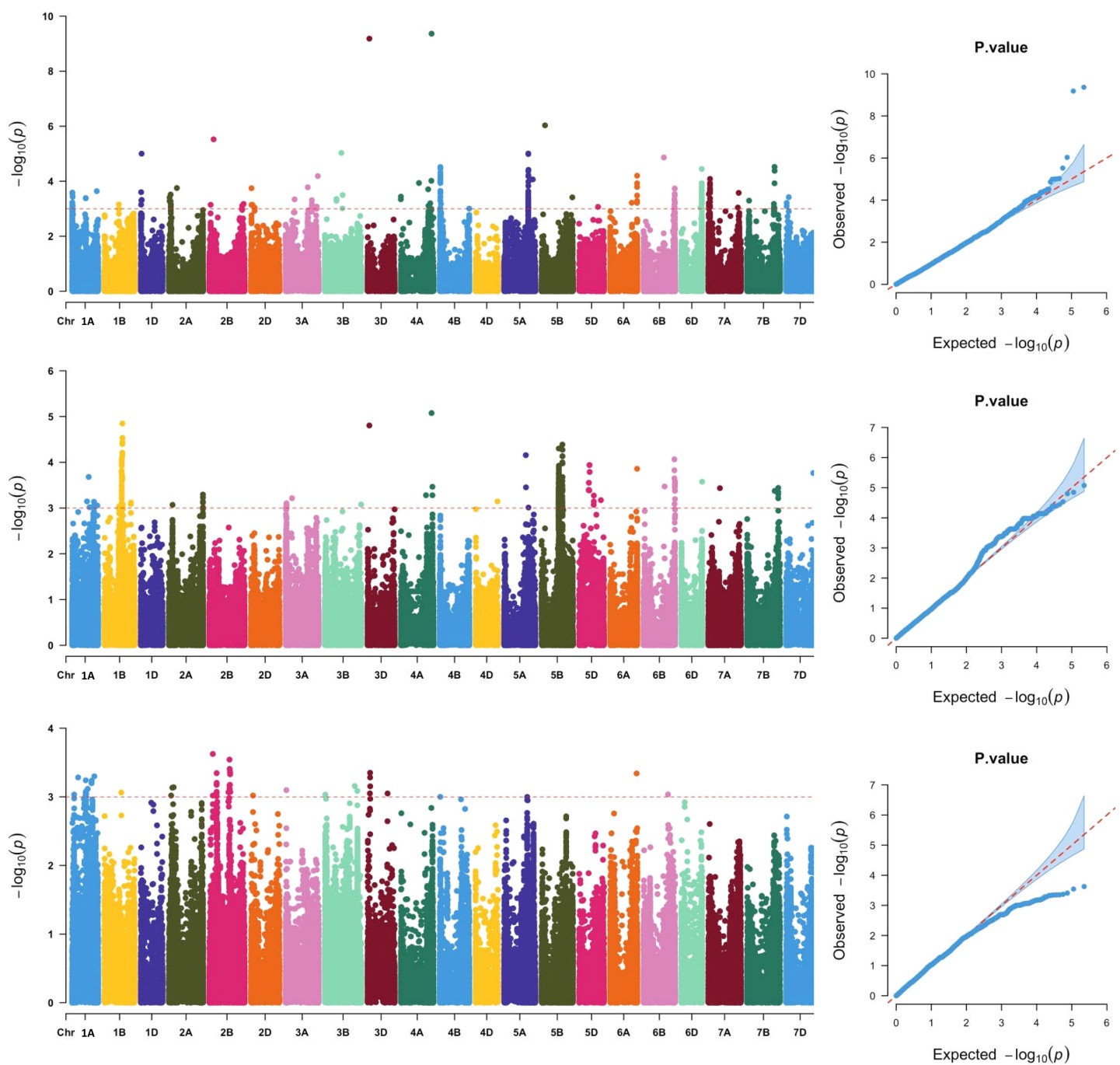

**Figure 2 Manhattan and QQ plots of GWAS for wheat grains Fe concentration in the wheat association mapping population based on the mixed linear model (MLM).** The horizontal red color line indicated the genome-wide significant threshold of $-\log_{10}$ (P-value) of 3.0. The SNPs above the red dotted line are significantly associated with grain Fe variation. Quantile-quantile scale representing expected *vs* observed $-\log10$ (P-value) in three environments (Kaifeng, Shangqiu and Yuanyang).

## Genome-wide association analysis of wheat grain Fe concentration

Across the three locations, a total of 911 SNPs were significantly associated with grain Fe concentration (Fig. 2, Table S2), which were distributed across all 21 chromosomes.

The phenotypic variation explained by each SNP ranged from 5.79% to 25.31%, suggesting that SNPs with moderate and minor effects on grain Fe concentration were detected. Association analysis of grain Fe concentration in three locations was further analyzed. In Yuanyang, there were 48 significant SNPs on chromosomes 2A, 2B, 2D, 3A, 3B, 3D, 6A, and 6B. However, in Kaifeng, 209 significant SNPs were detected, which were distributed on chromosomes 1D, 2A, 2B, 2D, 3A, 3B, 3D, 4A, 4B, 5A, 5B, 5D, 6A, 6B, 6D, 7A, 7B, and 7D. Compared with Yuanyang and Kaifeng, a greater number of significant SNPs were identified in Shangqiu. Although a total of 446 SNPs were identified in Shangqiu, each SNP had a lower explanation percentage for phenotypic variation. These SNPs were located on chromosomes 2A, 3A, 3D, 4A, 4D, 5A, 5B, 5D, 6A, 6B, 6D, 7A, 7B, and 7D. In addition, SNP number varied greatly across the different chromosomes, of which the highest number of SNPs was found on chromosome 5B (381) and the lowest on chromosome 4D and 6D (1), suggesting that the 5B chromosome is the main genetic region for grain Fe concentration. According to the flanking intervals of SNP, the identified SNPs could be categorized into 46 non-redundant QTLs (Table 2). The number of SNPs in each non-redundant QTL covered was different and varied from 1 to 64 SNPs. The number of QTLs varied across different genomes, and the highest number of QTLs was found in the A genome (18), followed by the B genome (16), while only 12 QTLs were identified in the D genome.

Significantly, two major SNPs were found on chromosomes 4A and 3D. One SNP, AX-108912427, was located at 699,571,654 bp on chromosome 4A. It explained 25.31% of the observed variation in grain Fe concentration. Another SNP, AX-94729264, could explain 24.84% of the observed phenotypic variations and was mapped to the physical position of 40,526,440 bp on chromosome 3D. In GWAS analysis, reliable SNPs that were simultaneously detected in more than two environments were considered more relevant for inbreeding the new varieties with high grain Fe concentration. In the present study, nine reliable SNPs, including AX-94729264, AX-108912427, AX-94936962, AX-109956643, AX-111493816, AX-111088162, AX-109899864, AX-94702817, and AX-95210102, were detected in Kaifeng and Shangqiu environments. Notably, the two significant SNPs (AX-108912427 and AX-94729264) not only have a more significant effect on grain Fe concentration but are also reliable under the different environments, so further study is needed for the two SNPs.

## Prediction of candidate genes

To understand the molecular mechanisms of Fe accumulation in wheat, candidate gene analysis was conducted for Fe. Candidate genes were predicted for the SNPs that were identified in GWAS analysis. An expression heat map was constructed for these candidate genes using the public database of Wheat Expression Browser (http://www.wheat-expression.com), and genes that were only specifically expressed in grain tissues were predicted as candidate genes for grain Fe concentration. Ten candidate genes were identified for grain Fe concentration (Table 3). On chromosome 3D, two candidate genes were found. One was TraesCS3D01G078500 that encoded for a NAC domain-containing protein, which showed potential relevance to metal remobilization and

**Table 2 List of significant loci and their detailed information for Fe concentration identified by GWAS.**

| Nu | Chromosome | Interval range | No. of SNPs | Environment | Peak SNP | Position | P Value | R²(%) |
|----|-----------|----------------|-------------|-------------|----------|----------|---------|-------|
| 1 | 1A | 12,192,544–504,855,880 | 35 | YY | AX-110912741 | 504,855,880 | $5.02 \times 10^{-4}$ | 12.19 |
| 2 | 1A | 3,260,219–560,890,453 | 17 | KF | AX-109487734 | 560,890,453 | $2.29 \times 10^{-4}$ | 11.66 |
| 3 | 1A | 338,830,974–574,937,064 | 15 | SQ | AX-95178074 | 377,027,087 | $2.05 \times 10^{-4}$ | 7.32 |
| 4 | 1B | 328,824,347–328,880,060 | 5 | KF | AX-111580083 | 328,879,038 | $6.97 \times 10^{-4}$ | 10.62 |
| 5 | 1B | 385,161,579 | 1 | YY | AX-94632727 | 385,161,579 | $8.66 \times 10^{-4}$ | 11.69 |
| 6 | 1B | 336,787,649–603,147,101 | 136 | SQ | AX-108903980 | 409,302,131 | $1.42 \times 10^{-5}$ | 10.06 |
| 7 | 1D | 5,421,805–20,056,510 | 5 | KF | AX-110529533 | 16,132,987 | $9.97 \times 10^{-6}$ | 14.71 |
| 8 | 2A | 17,032,001–182,302,531 | 10 | KF | AX-95247517 | 182,302,531 | $1.77 \times 10^{-4}$ | 11.91 |
| 9 | 2A | 52,130,032–108,517,228 | 4 | YY | AX-108896742 | 105,628,630 | $7.26 \times 10^{-4}$ | 11.85 |
| 10 | 2A | 83,586,426–775,041,568 | 6 | SQ | AX-111455865 | 775,029,238 | $5.07 \times 10^{-4}$ | 6.45 |
| 11 | 2B | 27,522,279–777,308,839 | 4 | KF | AX-110970921 | 95,738,751 | $3.01 \times 10^{-6}$ | 15.91 |
| 12 | 2B | 77,961,365–475,608,029 | 32 | YY | AX-108855338 | 77,961,365 | $2.37 \times 10^{-4}$ | 12.89 |
| 13 | 2D | 15,442,231–86,079,836 | 5 | KF | AX-111611520 | 15,442,231 | $1.79 \times 10^{-4}$ | 11.89 |
| 14 | 2D | 50,470,803 | 1 | YY | AX-95104146 | 50,470,803 | $9.58 \times 10^{-4}$ | 11.60 |
| 15 | 3A | 16,070,421 | 1 | YY | AX-108847051 | 16,070,421 | $7.98 \times 10^{-4}$ | 11.77 |
| 16 | 3A | 15,348,646–144,473,416 | 9 | SQ | AX-94863805 | 144,473,416 | $6.11 \times 10^{-4}$ | 6.26 |
| 17 | 3A | 202,587,965–731,220,289 | 13 | KF | AX-94987631 | 731,220,289 | $5.01 \times 10^{-5}$ | 12.87 |
| 18 | 3B | 10,301,481–737,764,644 | 3 | YY | AX-108861017 | 679,198,055 | $6.95 \times 10^{-4}$ | 11.89 |
| 19 | 3B | 263,249,368–404,801,032 | 7 | KF | AX-110922471 | 376,625,452 | $9.37 \times 10^{-6}$ | 14.77 |
| 20 | 3B | 825,341,801 | 1 | SQ | AX-110526198 | 825,341,801 | $8.37 \times 10^{-4}$ | 5.96 |
| 21 | 3D | 40,526,440 | 1 | KF | AX-94729264 | 40,526,440 | $5.84 \times 10^{-10}$ | 24.84 |
| 22 | 3D | 40,526,440 | 1 | SQ | AX-94729264 | 40,526,440 | $1.57 \times 10^{-5}$ | 9.95 |
| 23 | 3D | 52,650,282–454,744,784 | 5 | YY | AX-108814800 | 57,034,968 | $4.46 \times 10^{-4}$ | 12.30 |
| 24 | 4A | 451,704–699,571,654 | 8 | KF | AX-108912427 | 699,571,654 | $4.35 \times 10^{-10}$ | 25.31 |
| 25 | 4A | 572,558,552–718,836,253 | 4 | SQ | AX-108912427 | 699,571,654 | $8.41 \times 10^{-6}$ | 10.60 |
| 26 | 4B | 11,987,531–670,399,775 | 26 | KF | AX-111479123 | 14,490,156 | $3.06 \times 10^{-5}$ | 13.60 |
| 27 | 4D | 500,023,674 | 1 | SQ | AX-109847855 | 500,023,674 | $7.17 \times 10^{-4}$ | 6.11 |
| 28 | 5A | 492,890,942–553,064,045 | 4 | SQ | AX-109623019 | 492,890,942 | $6.99 \times 10^{-5}$ | 8.42 |
| 29 | 5A | 547,505,636–650,240,330 | 39 | KF | AX-109379942 | 549,160,416 | $9.86 \times 10^{-6}$ | 14.72 |
| 30 | 5B | 78,708,064–699,558,049 | 2 | KF | AX-112289745 | 78,708,064 | $9.33 \times 10^{-7}$ | 17.09 |
| 31 | 5B | 387,308,892–477,630,916 | 379 | SQ | AX-111708401 | 473,871,199 | $4.09 \times 10^{-5}$ | 8.97 |
| 32 | 5D | 218,945,113–496,411,812 | 12 | SQ | AX-89438182 | 238,177,337 | $1.15 \times 10^{-4}$ | 7.92 |
| 33 | 5D | 431,203,273 | 1 | KF | AX-95075283 | 431,203,273 | $8.57 \times 10^{-4}$ | 10.43 |
| 34 | 6A | 611,858,549 | 1 | SQ | AX-94936962 | 611,858,549 | $1.39 \times 10^{-4}$ | 7.73 |
| 35 | 6A | 601,398,500 | 1 | YY | AX-109557802 | 601,398,500 | $4.56 \times 10^{-4}$ | 12.28 |
| 36 | 6A | 490,486,081–612,124,269 | 8 | KF | AX-94975608 | 612,123,809 | $6.31 \times 10^{-5}$ | 12.90 |
| 37 | 6B | 556,751,602 | 1 | YY | AX-110421368 | 556,751,602 | $9.21 \times 10^{-4}$ | 11.64 |
| 38 | 6B | 462,555,585–712,475,640 | 48 | KF | AX-111084964 | 462,555,585 | $1.37 \times 10^{-5}$ | 14.39 |
| 39 | 6B | 479,332,942–716,010,197 | 17 | SQ | AX-95210102 | 708,943,119 | $3.04 \times 10^{-4}$ | 9.52 |
| 40 | 6D | 471,071,095 | 1 | SQ | AX-94688853 | 471,071,095 | $2.66 \times 10^{-4}$ | 7.08 |
| 41 | 7A | 261,687,749 | 1 | SQ | AX-111012263 | 261,687,749 | $3.67 \times 10^{-4}$ | 6.76 |

(Continued)

| Nu | Chromosome | Interval range | No. of SNPs | Environment | Peak SNP | Position | P Value | R²(%) |
|---|---|---|---|---|---|---|---|---|
| 42 | 7A | 11,099,182–688,980,095 | 21 | KF | AX-109282301 | 36,334,168 | $8.23 \times 10^{-5}$ | 12.65 |
| 43 | 7B | 47,617,941–627,982,714 | 8 | KF | AX-111148246 | 626,954,936 | $3.01 \times 10^{-5}$ | 13.62 |
| 44 | 7B | 630,565,865–711,075,593 | 7 | SQ | AX-94839775 | 711,067,695 | $3.63 \times 10^{-4}$ | 6.77 |
| 45 | 7D | 11,666,920–58,784,583 | 3 | KF | AX-110930232 | 58,784,583 | $3.79 \times 10^{-4}$ | 11.19 |
| 46 | 7D | 616,767,211–616,767,235 | 2 | SQ | AX-94455581 | 616,767,235 | $1.72 \times 10^{-4}$ | 7.52 |

**Note:**
Nu refer to the number of the loci detected in this study; Peak SNP refer to the most significant SNP in the mapping interval; Position refer to the physical position of most significant SNPs in the mapping interval; P Value means P value of the target trait calculated by MLM model; R² refer to the percentage of phenotypic variance explained by the locus. KF, SQ, YY refer to Kaifeng, Shangqiu, Yuanyang environment, respectively.

**Table 3 The candidate genes and their information for grain Fe concentration identified in this study.**

| Nu[a] | Chromosome | Identified loci in current study | Position (bp)[b] | Candidate genes (closest/nearby) | Annotation |
|---|---|---|---|---|---|
| 1 | 3D | AX-94729264 | 40,526,440 | TraesCS3D01G078500 | NAC domain-containing protein |
| 2 | | | | TraesCS3D01G080900 | defensin-like protein |
| 3 | 4A | AX-108912427 | 699,571,654 | TraesCS4A01G430000 | DUF581 family protein |
| 4 | | | | TraesCS4A01G431200 | Acid phosphatase 1 |
| 5 | | | | TraesCS4A01G431800 | senescence-associated family protein, putative (DUF581) |
| 6 | | | | TraesCS4A01G431900 | senescence-associated family protein, putative (DUF581) |
| 7 | | | | TraesCS4A01G432000 | DUF581 family protein |
| 8 | 6A | AX-94936962 | 611,858,549 | TraesCS6A01G403500 | Remorin |
| 9 | 6B | AX-94702817 | 708,943,077 | TraesCS6B01G447400 | Remorin |
| 10 | | | | TraesCS6B01G449700 | Ring finger protein, putative |

**Notes:**
[a] The number of candidate genes for wheat grain Fe concentration.
[b] Physical position of the SNP as reported in the IWGSC Chinese Spring reference genome RefSeq v2.0.

accumulation in wheat. Another gene is TraesCS3D01G080900, which encodes a defensin-like protein that has biological activities in ion channel blockage. Five candidate genes, namely TraesCS4A01G430000, TraesCS4A01G431200, TraesCS4A01G431800, TraesCS4A01G431900, and TraesCS4A01G432000,were found on chromosome 4A. Interestingly, TraesCS4A01G430000 and TraesCS4A01G432000encode plant unknown function DUF581 family proteins that possibly participate in mineral translocation to seeds. However, TraesCS4A01G431800 and TraesCS4A01G431900 encoded an associated family protein (DUF581) that is related to senescence and participates in transporting Fe ions. Only one gene, TraesCS4A01G431200, was associated with acid phosphatase. Acid phosphatase plays a role in providing the desired energy for active transport and possibly affects Fe uptake and transport. TraesCS6A01G403500 on chromosome 6A encodes the remorin protein that may be associated with the cytoskeleton or membrane skeleton and may play a role in regulating Fe translocation, while TraesCS6B01G449700 on chromosome 6B encodes a ring finger protein that is relevant to the accumulation of more Fe in grains.

**Table 4 Variance analysis for haplotypes with different alleles and haplotype combinations in two blocks on chromosome 6B.**

| Block | Haplotypes | Fe_KF | Fe_SQ | Fe_YY | Hap frequency (%) |
|---|---|---|---|---|---|
| Block1 | Hap1A | 98.35 ± 6.52 | 67.32 ± 2.23 | 63.32 ± 2.44 | 34.00 |
| | Hap1B | 102.06 ± 5.83 | 73.45 ± 1.88* | 70.59 ± 2.19* | 66.00 |
| Block2 | Hap2A | 92.90 ± 6.02 | 68.19 ± 2.32 | 64.28 ± 2.65 | 70.00 |
| | Hap2B | 96.41 ± 10.18 | 77.27 ± 3.78* | 74.20 ± 5.20 | 30.00 |
| Block1+ Block2 | Hap1A+Hap2A | 92.90 ± 6.02 | 68.19 ± 2.32 | 64.28 ± 2.65 | 70.73 |
| | Hap1B+Hap2B | 99.48 ± 10.15 | 76.91 ± 3.94* | 75.49 ± 5.25* | 29.27 |

**Note:**
* Significant differences between haplotype A and haplotype B ($P < 0.05$). Fe_KF, Fe_SQ, Fe_YY refer to grain-Fe content of haplotypes in Kaifeng, Shangqiu and Yuanyang environment, respectively.

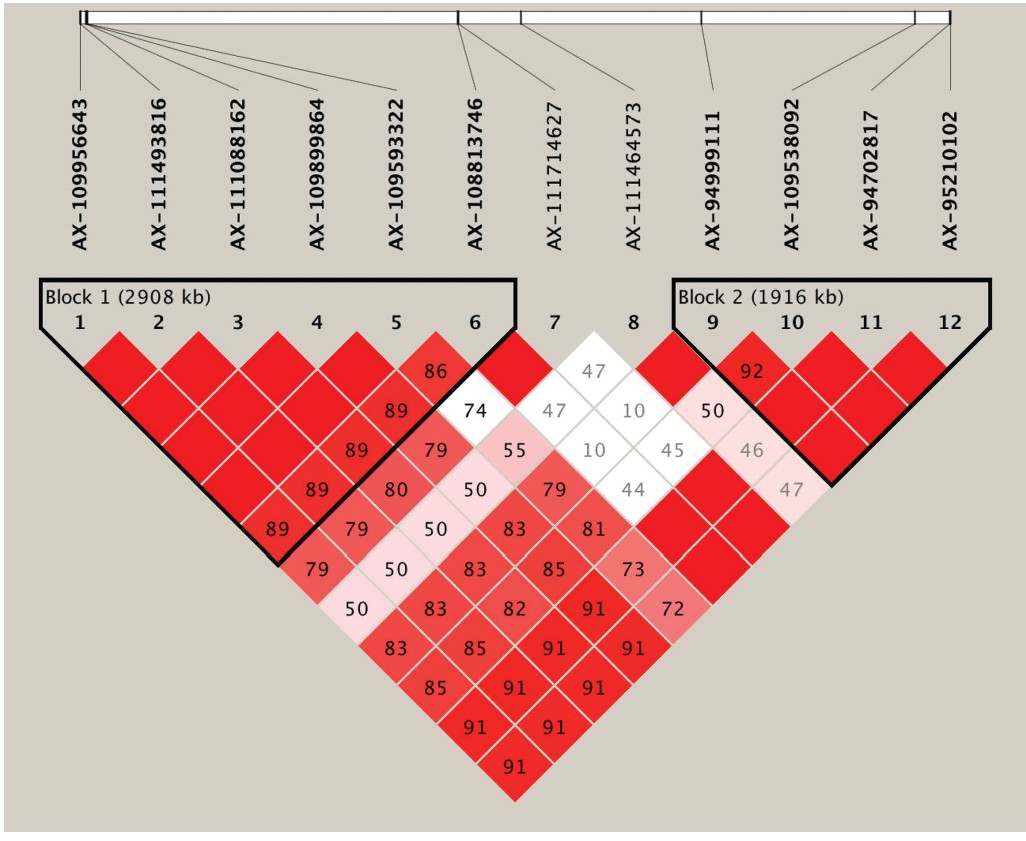

**Figure 3 Haplotype analysis for the significant SNPs associated with wheat grains Fe concentration on chromosome 6B.** Haplotype heatmap surrounding significant SNPs on chromosome 6B.

## Haplotypes associated with grain Fe concentration

Based on the genome-wide association analysis, we detected a significant cluster of 12 SNPs on chromosome 6B. In the corresponding region, 10 SNPs exhibited strong LD and could form two LD blocks (block1 and block2) (Table 4, Fig. 3, Table S3). The first large LD block spanned approximately 2,908 kb, including six SNPs with two haplotypes (Hap1A and Hap1B), and the second LD block spanned approximately 1,916 kb, including

four SNPs with two possible haplotypes (Hap2A and Hap2B). To compare the effect of different haplotypes, we analyzed the effect of each haplotype on grain Fe concentration variation across the three environments. The variation of grain Fe concentration in Hap1 all reached the significant levels in the YY and SQ environments, but in Hap2, the significant level for grain Fe concentration was only observed in the SQ environment. In the KF environment, although Hap1 or Hap2 had non-significant effects on the grain Fe concentrations, the mean of grain Fe concentration with HapB was higher than that of HapA, which coincided with the distribution tendency in YY and SQ. The combined effect of the Hap1 and Hap2 blocks was also analyzed. In the combination of Hap1+Hap2, the variations in grain Fe concentration reached significant levels in YY and SQ. In addition, the combination Hap1A+Hap2A had higher frequency (70.73%) compared to haplotype Hap1B+Hap2B (29.27%).

## Comparison with the previous studies

Several previous studies have been conducted for wheat grain Fe, and some QTLs were identified for this trait. We retrieved a total of 94 wheat Fe QTLs that were distributed among all 21 chromosomes except 1D (Table S4). To compare the Fe QTLs of this study with that of previous studies, the physical positions of Fe QTLs were determined using the closest linked markers. Ultimately, the physical position of 57 QTLs was successfully determined (*Tong et al., 2020*). When we compared Fe QTLs of this study with those of the previous investigations, major differences in QTL amount or QTL physical position were observed. In the present study, a total of 46 Fe QTLs were identified, whereas a total of 94 QTLs were found in all of the previous studies. We found that the position of a few QTLs of this study nearly coincided with that of the previous studies. For example, *Gorafi et al. (2016)* and our group found one wheat Fe QTL near the 496 Mb position on chromosome 5D, and *Pu et al. (2014)* and our team also found one wheat Fe QTL near the 616 Mb position on chromosome 7D. In addition, we found some QTLs in locations similar to those of their counterparts from previous studies, which suggests that at least some of the results of this study are consistent with those of previous studies (*Arora et al., 2019*; *Alomari et al., 2018*; *Crespo-Herrera et al., 2017*; *Cu et al., 2020*; *Rathan et al., 2021*; *Roshanzamir, Kordenaeej & Bostani, 2013*; *Wang et al., 2021a*; *Wang et al., 2021b*). No other QTLs reported in the present study were identified in previous studies. For instance, on chromosome 1D, we detected one QTL, but this QTL had not been documented elsewhere.

## DISCUSSION

Breeding elite wheat varieties with high microelement content has proven to be an effective biofortification method to address micronutrient malnutrition. The kernel is the most important organ of wheat, which provides useful information for biofortification breeding, and therefore has become the main study objective in Fe content research. Increasing micronutrient content in wheat through breeding requires the existence of substantial genetic variation for this trait. Selecting the accessions with high Fe grain concentration from the diverse germplasms is important in wheat breeding programs for enhancing Fe

content. Several previous studies have been conducted for detecting the variation of wheat grain Fe content (*Garnett & Graham, 2005*; *Xu et al., 2012*). The present study demonstrated that the association population contained the plentiful genetic variations in grain Fe concentration. The high Fe concentration in some wheat accessions, for example, KH438, KH445, and KH242, as the elite lines, could be used for developing new wheat varieties with high grain Fe concentration.

The understanding of the genetic basis of micronutrient accumulation in the wheat grain and mapping of the quantitative trait loci (QTL) will provide useful information for improving grain micronutrient concentrations through marker-assisted selection (MAS). The present associated mapping population exhibits a quantitative mode of inheritance for grain Fe concentration, which is in agreement with previous reports (*Tiwari et al., 2009*; *Srinivasa et al., 2014*). Numerous previous QTL mapping studies have been conducted for wheat grain Fe (*Arora et al., 2019*; *Alomari et al., 2018*; *Cu et al., 2020*; *Gorafi et al., 2016*; *Rathan et al., 2021*; *Velu et al., 2017*; *Wang et al., 2021a*; *Wang et al., 2021b*). A previous study conducted by *Pu et al. (2014)* reported five QTLs for grain Fe on chromosomes 2B, 4A, 5A, 5B, and 7D using recombinant inbred line population of wheat, and these QTLs coincided with the lociAX-111087936, AX-109551612, AX-110549899, AX-109486399, and AX-95112608 identified in this study. Notably, in our study, two major SNPs with the highest phenotypic variation explained valuewere identified for wheat grain-Fe concentration. One SNP (AX-108912427) was mapped to the near genomic region of one QTL on chromosome 4A reported by *Pu et al. (2014)*. Another major SNP (AX-94729264) located on 3D chromosome near to*QGFe.co-3D* was reported by *Liu et al. (2019)*. Using the two wheat RIL mapping populations, *Velu et al. (2017)* identified four QTLs for wheat grain Fe on chromosomes 2A, 2B, 5B, and 7B. In the near genomic regions of the four QTLs, AX-94513201, AX-109274500, AX-108921926, and AX-110024541 were found in our research. Interestingly, we observed that the loci for wheat grain Fe were seldom found on 1D chromosome previously. Until recently, *Rathan et al. (2021)* reported one QTL for wheat grain-Fe on chromosome 1D. In our research, we found five significant SNPs for wheat grain Fe on chromosome 1D, which suggested this is the first time these SNPs were reported to the best of our knowledge. In the current study, all significant SNPs are scattered across the entire wheat genome. In the comparatively narrow region on 6B chromosome, we detected a significant cluster of 12 SNPs, which spanned approximately 4,824 kb. This chromosome segment can be incorporated into breeding efforts in cultivating a high-Fe accumulation germplasm. Compared with previous studies, we detected more loci for wheat grain Fe concentration, possibly due to the high-density linkage map constructed with a large number of gene-based SNP markers based on a 660K SNP array in this study.

Identifying genes within a QTL region can help elucidate trait architecture if gene function can be related to the associated trait. Although several studies have reported QTLs for grain Fe concentration in wheat, only a few candidate genes have previously been identified. The genetic mechanisms of wheat grain Fe concentration are presently unclear. In this study, by candidate gene analysis, we found 10 candidate genes for wheat grain Fe concentration. The functions of these candidate genes are related to uptake, transport,

translocation, remobilization, and accumulation in wheat plants. Of the 10 candidate genes, TraesCS3D01G078500 encoded for a NAC domain-containing protein. *Ogo et al. (2008)* isolated an NAC transcription factor, IDEF2, from rice and barley and revealed IDEF2 as a key transcription factor regulating the Fe deficiency response. *Uauy et al. (2006)* also reported one ancestral wild wheat allele, Gpc-B1, that is associated with increased Fe content and encodes an NAC transcription factor that accelerates senescence and increases nutrient remobilization from leaves to developing grains. Interestingly, in the identified ten candidate genes, four genes were related to senescence. Nutrients, including most minerals, are mobilized from senescing leaves to other tissues during leaf senescence (*Himelblau & Amasino, 2001*). In wild-type wheat, Fe mobilization from the vegetative parts to the grain during leaf senescence and grain maturation appears to be quite efficient, so most of the total Fe is translocated to the grains (*Garnett & Graham, 2005*). The timing and efficiency of senescence-mediated nutrient mobilization appears to be a primary determinant for grain Fe content. Previous studies have indicated that the NAM-B1 gene, which encodes a member of the NAC transcription factor gene family, could play a central role in senescence and nutrient mobilization (*Uauy et al., 2006*). In this study, TraesCS4A01G430000, TraesCS4A01G432000, TraesCS4A01G431800, and TraesCS4A01G431900 encoded the associated family protein (DUF581). The four candidate genes are all related to senescence and transportation of Fe ions. On 3D chromosome, another gene, TraesCS3D01G080900, encodes a defensin-like protein that has biological activities involving ion channel blockage. More than 500 defensin proteins have been discovered to date. Most plant defensins have been isolated from seeds, and these have also been identified in vegetative tissues (*Gachomo et al., 2012*). TraesCS6A01G403500 encodes a remorin protein. Remorin has been found in detergent-insoluble membranes and may be associated with the cytoskeleton or membrane skeleton, which suggests that remorin may play a role in regulating Fe uptake.

Increasing grain-Fe content in wheat has received more attention in recent years and become an important quality breeding objective in wheat practice around the world. However, to date, the genetic mechanism controlling grain Fe content is still unclear in wheat. In the present study, a total of 911 significant SNPs associated with grain Fe concentration were identified, and 10 candidate genes were predicted. Collectively, these findings have three potential uses. First, these findings will provide some useful wheat germplasms with highly favorable alleles of Fe content for breedingthe elite Fe enrich varieties. Second, the markers identified in this study could be utilized in molecular marker-assisted breeding (MAS) for biofortification of wheat to increase kernel-Fe content. The last and perhaps most promising is the utilization of important regions with stacking significant SNPs, and candidate genes will facilitate the breeding of Fe-enriched wheat varieties. Because most of the significant SNPs and candidate genes in this study were not reported before in wheat, more further studies are needed to validate our findings in the future.

## CONCLUSIONS

Briefly, two major SNPs and nine reliable SNPs for wheat grain Fe were identified, and 10 candidate genes were predicted. All significant SNPs identified in this study scattered across the entire wheat genome, and one loci cluster was found on 6B chromosome. The functions of the candidate genes are primarily associated with transport, translocation, remobilization, and accumulation of Fe in wheat plants. Therefore, this study provides useful loci andgene information for improving Fe concentration in wheat grain. In the future, further studies need be conducted for the candidate genes identified herein, which would be helpful to elucidate the molecular mechanisms of Fe content in the wheat grain.

### Funding

This research was funded by the Joint Fund of National Natural Science Foundation of China (U1804102), the Scientific and Technological Research Project of Henan Province (202102110027, 212102110065), the Excellent Youth Fund of Henan Academy of Agricultural Sciences (2020YQ02, 2022YQ13), and the China Postdoctoral Science Foundation (2020M682300). The funders had no role in study design, data collection and analysis, decision to publish, or preparation of the manuscript.

### Grant Disclosures

The following grant information was disclosed by the authors:
National Natural Science Foundation of China: U1804102.
Scientific and Technological Research Project of Henan Province: 202102110027, 212102110065.
Henan Academy of Agricultural Sciences: 2020YQ02, 2022YQ13.
China Postdoctoral Science Foundation: 2020M682300.

### Competing Interests

The authors declare that they have no competing interests.

### Author Contributions

- Jiansheng Wang conceived and designed the experiments, performed the experiments, analyzed the data, prepared figures and/or tables, authored or reviewed drafts of the article, and approved the final draft.
- Xia Shi analyzed the data, prepared figures and/or tables, and approved the final draft.
- Zhengfu Zhou analyzed the data, authored or reviewed drafts of the article, and approved the final draft.
- Maomao Qin performed the experiments, authored or reviewed drafts of the article, and approved the final draft.
- Yahuan Wang performed the experiments, authored or reviewed drafts of the article, and approved the final draft.
- Wenxu Li performed the experiments, authored or reviewed drafts of the article, and approved the final draft.
- Pan Yang performed the experiments, authored or reviewed drafts of the article, and approved the final draft.
- Zhengqing Wu analyzed the data, authored or reviewed drafts of the article, and approved the final draft.
- Zhensheng Lei conceived and designed the experiments, authored or reviewed drafts of the article, and approved the final draft.

## Data Availability

The raw data is available in the Supplemental Files.

## Supplemental Information

Supplemental information for this article can be found online at http://dx.doi.org/10.7717/peerj.13625#supplemental-information.

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
