# Peer review of "Genetic dissection of grain iron concentration in hexaploid wheat (Triticum aestivum L.) using a genome-wide association analysis method"

_PeerJ, doi:10.7717/peerj.13625_

## Round 0.1 · original submission · Major Revisions

Dear Dr. Wang:

Thank you for your submission to PeerJ.

Your manuscript has been reviewed by two experts in your research area. Based on their reviews and my assessment, your manuscript entitled "Genetic dissection of grain iron concentration in hexaploid wheat (Triticum aestivum L.) using a genome-wide association analysis method" requires Major Revisions. Thank you,

Best regards,

Sincerely,

Tika Adhikari

Reviewer 1 ·

Basic reporting

A sufficient field background was provided. However, Some of the sentences need to be revised (mentioned in the article).
Figures are relevant, however, Figures 2, 3, 4 could be merged into a single one. Similarly, Figures of Manhattan plots (Figures 5, 6, 7) and QQ plots (Figures 8, 9, 10) could be merged into a single one. It would be better to have LD and population structure figures
In Tables, some of the missing information must be included (mentioned in the article)

Experimental design

The research was well designed, relevant information was provided and the research was conducted in the prevailing ethical standards

Validity of the findings

The results were well written, however, some of the sections could be revised (mentioned in the article).
Some of the sentences are repeated and the flow of the information is lacking in the discussion section.
The conclusion needs to be revised

Additional comments

NA

Annotated reviews are not available for download in order to protect the identity of reviewers who chose to remain anonymous.

·

Basic reporting

This article was well described, and the results will be available for improving the wheat Fe content by MAS in future. But it needs major revision. Detail revisions were as follows:

Introduction
(1) In this part, the advantages of GWAS and SNP marker were introduced too much, but there is no introduction on previous researches on mapping of minerals using linkage map and GWAS. So recommend the author to add this content.
(2) Line 85: “of” should be deleted.

Materials and methods

Line134: “ Heinongjiang” changed to "Heilongjiang"
Line140-143 and Line 145-146: the wheat grains were physiologically mature or completely mature for Fe content analysis? It seemed to be inconsistent.
Line166 and Line169: the number of SNP markers “244,508” or “1.25 million SNPs” in GWAS analysis was inconsistent.
Line187-188, this sentence is not clear.

In addition, because the Fe content of grain is affected by soil minerals, so the minerals content of soil at three experiment locations should be introduced.

Results
Line191-194: The sentences from “Grain Fe concentration” to “depicted in Figure 2, Figure 3 and Figure 4.” Should be deleted.
Line213 "." should be deleted
Line272 “Detailed descriptions of the 10 candidate genes are summarized in Table 3.” Should be deleted.
Line301 “and” should be added before “Pu”.

The references in latest three years should be added and compared analysis, for example, the reference “Identification of novel genomic regions associated with nine mineral elements in Chinese winter wheat grain” has been published on BMC Plant Biol (2021) 21:311”.

Discussion needs further revision.

Experimental design

Because the Fe content of grain is affected by soil minerals, so the minerals content of soil at three experiment locations should be introduced.

Validity of the findings

The results will be available for improving the wheat Fe content by MAS in future.

Additional comments

Major revision

---

## Round 0.2 · Minor Revisions

Dear Dr. Wang:

Thank you for your patience and understanding. Please see the attached file and revise your manuscript. Thank you.

Best regards,

Tika Adhikari

Reviewer 1 ·

Basic reporting

A sufficient field background was provided. The authors have changed the article however, some of the sentences need to be revised (mentioned in the article).

Experimental design

The research was well designed, relevant information was provided and the research was conducted in the prevailing ethical standards

Validity of the findings

The authors have changed the article however, Some of the sections could be revised (mentioned in the article).

Annotated reviews are not available for download in order to protect the identity of reviewers who chose to remain anonymous.

·

Basic reporting

After revision, it has improved, but still needs to be revised in tables and figures. Details are as follows:
In table1, there were no explaination for the three letters,"h,j,j" , please deleted or explaination.
In Figure2, the 21 chromosomes should be added at the Horizontal axis.
In table2, in the coloum P value, the numbers should be superscripted, please check the number.

Experimental design

no comment

Validity of the findings

no comment

Additional comments

no

---

## Round 0.3 · Minor Revisions

Dear authors:

Thank you very much for revising your manuscript entitled "Genetic dissection of grain iron concentration in hexaploid wheat (Triticum aestivum L.) using a genome-wide association analysis method".
Also, thank you for your patience and understanding of the delay in the review process of your manuscript.

I had received a few comments (below) from the Section Editor and would highly appreciate your revising the manuscript before accepting it for publication in PeerJ.

Comments:

1: p-value threshold of 0.001 is way more lenient than the standard procedure in GWAS and is not appropriate. Please note that at p < 0.001 one expects a false positive in 1 out of every 1,000 tests by random chance. 200,000 + SNPs are tested here, so at p < 0.001 we expect 2,000 false positives. Possible approaches are Bonferroni, FDR, or an FDR that tries to account for LD among SNPs. Please check this reference, https://doi.org/10.1534/genetics.116.193987.

2. GWAS methods are not adequately described. GAPIT has many models, just saying that GAPIT was used with a kinship matrix is not enough. Which model was used?

3. Statistical analysis likely should be improved. Specifically, a mixed-effect model accounting for blocks should be used.

4. Broad sense heritability needs to be calculated. \

5. Manuscript needs significant editing for proper grammar."

Thank you.

Best regards,

Tika Adhikari

---

## Round 0.4 · accepted · Accept

Dear Dr. Wang,

Thank you for revising and submitting your manuscript to PeerJ.

I am writing to inform you that your manuscript - Genetic dissection of grain iron concentration in hexaploid wheat (Triticum aestivum L.) using a genome-wide association analysis method - has been accepted for publication. Congratulations!

Sincerely,

Tika Adhikari

Please see the attached file from the Academic Editor.